# Protocol for an independent patient data meta-analysis of prophylactic mesh placement for incisional hernia prevention after abdominal aortic aneurysm surgery: a collaborative European Hernia Society project (I-PREVENT-AAA)

Rudolf van den Berg [1,2] Floris P J den Hartog,[1] Christina Bali,[3] Miltiadis Matsagkas,[4] Paul M Bevis,[5] Jonothan J Earnshaw,[6] Eike S Debus,[7] Susanne Honig,[8] Frederik Berrevoet,[9] Olivier Detry,[10] Cesare Stabilini,[11] Filip Muysoms,[12] Pieter J Tanis,[1] European Hernia Society Prophylactic mesh study group collaborators

**To cite:** van den Berg R, den Hartog FPJ, Bali C, *et al.* Protocol for an independent patient data meta-analysis of prophylactic mesh placement for incisional hernia prevention after abdominal aortic aneurysm surgery: a collaborative European Hernia Society project (I-PREVENT-AAA). *BMJ Open* 2024;**14**:e081046. doi:10.1136/ bmjopen-2023-081046

For numbered affiliations see end of article.

**Correspondence to**
Rudolf van den Berg;
r.vandenberg.4@erasmusmc.nl

## ABSTRACT

**Introduction** Incisional hernia (IH) is a prevalent and potentially dangerous complication of abdominal surgery, especially in high-risk groups. Mesh reinforcement of the abdominal wall has been studied as a potential intervention to prevent IHs. Randomised controlled trials (RCTs) have demonstrated that prophylactic mesh reinforcement after abdominal surgery, in general, is effective and safe. In patients with abdominal aortic aneurysm (AAA), prophylactic mesh reinforcement after open repair has not yet been recommended in official guidelines, because of relatively small sample sizes in individual trials. Furthermore, the identification of subgroups that benefit most from prophylactic mesh placement requires larger patient numbers. Our primary aim is to evaluate the efficacy and effectiveness of the use of a prophylactic mesh after open AAA surgery to prevent IH by performing an individual patient data meta-analysis (IPDMA). Secondary aims include the evaluation of postoperative complications, pain and quality of life, and the identification of potential subgroups that benefit most from prophylactic mesh reinforcement.

**Methods and analysis** We will conduct a systematic review to identify RCTs that study prophylactic mesh placement after open AAA surgery. Cochrane Central Register of Controlled Trials, MEDLINE Ovid, Embase, Web of Science Core Collection and Google Scholar will be searched from the date of inception onwards. RCTs must directly compare primary sutured closure with mesh closure in adult patients who undergo open AAA surgery. Lead authors of eligible studies will be asked to share individual participant data (IPD). The risk of bias (ROB) for each included study will be assessed using the Cochrane ROB tool. An IPDMA will be performed to evaluate the efficacy, with the IH rate as the primary outcome.

## STRENGTHS AND LIMITATIONS OF THIS STUDY

⇒ We designed our protocol in collaboration with the European Hernia Society, an internationally recognised organisation with experience in procedures for navigating the safe transfer and storage of individual patient data (IPD).

⇒ IPD meta-analyses of randomised clinical trials enhance the ability to handle participant-level and study-level confounding factors and increase the power to identify responder subgroups and confounding factors underlying treatment effects.

⇒ A key limitation to undertaking IPD analyses relates to overcoming data-sharing hurdles, and the achievement of our aims will in part depend on the ability to successfully obtain IPD from eligible studies.

⇒ The protocol for this independent patient data meta-analysis was written according to the Preferred Reporting Items for Systematic Reviews and Meta-Analyses Protocol guidelines.

Any signs of heterogeneity will be evaluated by Forest plots. Time-to-event analyses are performed using Cox regression analysis to evaluate risk factors.

**Ethics and dissemination** No new data will be collected in this study. We will adhere to institutional, national and international regulations regarding the secure and confidential sharing of IPD, addressing ethics as indicated. We will disseminate findings via international conferences, open-source publications in peer-reviewed journals and summaries posted online.

**PROSPERO registration number** CRD42022347881.

## INTRODUCTION

Incisional hernia (IH) is a type of ventral abdominal wall hernia, which occurs in or near the scar of a previous surgical incision. The typical presentation is a visible or palpable bulge, which increases in size and visibility when the intra-abdominal pressure (IAP) is raised. Patients with IH are at risk for incarceration, bowel obstruction or strangulation, with an ischaemic bowel and emergency surgery with potential bowel resection and ostomy formation as a result.[1 2] Patients' daily functioning and social life can be affected, and serious mental issues can arise due to a changed body image.[3–5] IH repair has a big economic burden due to its prevalence and costs.[6] The only curative therapy is surgical reconstruction with mesh implantation, which can be very extensive surgery depending on hernia characteristics such as the diameter and location of the hernia.

Patients who undergo elective abdominal or pelvic surgery, where a median laparotomy is performed, have an up to 30% risk of IH formation. Typically, IH becomes evident within 2 years postoperatively.[7–9] In high-risk groups or after emergency surgery, the IH incidence can become as high as 69%.[10–14] High-risk groups are patients with a high body mass index (BMI, $\geq 27\,\mathrm{kg/m^2}$) or patients who underwent open repair of an abdominal aortic aneurysm (AAA).[15] Patients with an AAA might have an underlying connective tissue disorder, and it is hypothesised that this impairment also plays a role in the pathogenesis of IH.

Prevention of IH formation is a key issue in abdominal wall research. Different incision directions and locations, suture techniques and prophylactic reinforcement with mesh have been considered, with mixed outcomes. Conventional meta-analyses (MA) of randomised controlled trials (RCTs) have demonstrated that, in general, prophylactic mesh augmentation (PMA) after midline laparotomy is effective, safe and cost-effective.[16] However, due to problems with study design and sample size, the strength of recommendations for actually incorporating PMA in daily practice for elective midline laparotomies is weak.[16] PMA has also been studied in high-risk groups, although in a much smaller number of studies.[3] For AAA specifically, the European Society for Vascular Surgery guideline states that PMA after open AAA repair 'may be considered' (a class IIb recommendation, level of evidence A).[14] This recommendation is based on one of the latest MA (table 1).[17] Long-term results of two RCTs in that analysis, the PRIMA and PRIMAAT trials, have not yet been included in any MA.[18 19]

To date, no study on this topic has pooled individual participant data (IPD) across studies. An IPD meta-analysis (IPDMA) evaluates raw units of data rather than aggregated study-level data and is thus a more robust approach to evaluating treatment effect modifiers and mediators. Compared with traditional study-level MAs, IPDMAs enhance the ability to handle participant-level and study-level confounding factors, provide more complete analyses of time-to-event outcomes and increase the power to identify responder subgroups and mechanisms underlying the treatment effects. The outcomes resulting from using such an approach may, therefore, be more reliable and generalisable.

By combining the IPD of relevant RCTs together and performing statistical analyses on the combined, patient-level data, we strive to raise the level of evidence regarding mesh prophylaxis for IH prevention after open AAA repair and to help identify those who will benefit most from this procedure. This can only be achieved through international collaboration. Despite the growing recognition of the ethical and scientific importance of data sharing and scientific transparency, one of the biggest challenges in undertaking IPD analyses relates to overcoming data-sharing hurdles. Barriers range from successfully reaching original study authors, willingness or ability of authors to share data and international ethics and regulations issues. For this study, the collaboration will be initiated through the European Hernia Society (EHS), an internationally recognised organisation in the field of hernia surgery, and it has appointed a steering committee to oversee this IPDMA.

### Aims

We aim to conduct a systematic review and IPDMA of RCTs, to evaluate the effectiveness of the use of a prophylactic mesh after open AAA surgery as compared with primary sutured closure with IH rate during long-term follow-up (2-year, 3-year and 5-year IH rates) as primary outcome. Our secondary aims are to evaluate differences

**Table 1** Most recently published summary data of incisional hernia prevention by prophylactic mesh placement

| Study | Types of surgery | Risk ratio of IH incidence | Risk ratio of reoperation for IH | Risk-ratio postoperative seroma | Risk ratio postoperative SSI |
|---|---|---|---|---|---|
| Indrakusuma et al[17] (2018) | AAA open repair surgery | 0.27 (0.11–0.66) | 0.23 (0.05–1.05) | x | x |
| Aiolfi et al[21] (2023) | All midline incisions | 0.38 (0.24–0.58) | x | 2.05 (1.35–3.13) | 1.17 (0.82–1.67) |
| Jairam et al[3] (2020) | All elective midline incisions | 0.35 (0.21–0.57) | x | Onlay 2.23 (1.10–4.52) Retromuscular 1.67 (0.81–3.47) | Onlay 0.82 (0.55-1.23) Retromuscular 0.85 (0.50–1.45) |

AAA, abdominal aortic aneurysm; IH, incisional hernia; SSI, surgical site infection.

in postoperative complications within 30 days such as surgical site infection (SSI), surgical site occurrence (SSO) and fascial dehiscence, as well as pain (eg, visual analogue scale (VAS) pain score, numeric rating scale (NRS) pain score), quality of life (eg, EQ-4D, SF-36) and the need for re-operation (abdominal-wall and other) at different time points during follow-up (eg, <30 days, 6 months, 1 year). Furthermore, we aim to identify potential subgroups of patients who will benefit most from prophylactic mesh reinforcement after open AAA surgery regarding the reduction in IH rate. The results of this study are assumed to support recommendations in future guideline updates, and they will directly inform clinicians regarding the type of abdominal wall closure after open AAA repair. This will translate into benefits for those who will undergo AAA repair. Ultimately, reducing the incidence of IH after AAA repair is a socially responsible goal, as it will also result in reduced societal healthcare costs.

## METHODS AND ANALYSIS

The basic study protocol was approved by the EHS scientific committee. Subsequently, it was submitted for registration to the International Prospective Register of Systematic Reviews (PROSPERO) (registration number CRD42022347881). It formed the basis for the present, detailed protocol, which was written in accordance with the Preferred Reporting Items for Systematic Reviews and Meta-Analyses Protocols (PRISMA-P) statement and PRISMA-IPD guidelines. Data transfer methods, developed in collaboration with the Erasmus MC data transfer office (DTO) and approved by the EHS, will guide the secure transfer and responsible use of IPD, adhering to current European data-sharing regulations.

### Study identification

A literature search will be performed in the following databases: Cochrane Central Register of Controlled Trials (CENTRAL), MEDLINE Ovid (1946 onwards), Embase (1980 onwards), Web of Science Core Collection (1975 onwards) and Google Scholar. The search strategy will be tailor-made, by the investigators, together with an experienced, professional librarian from the Erasmus MC Medical Library. The complete search terms are noted in the online supplemental file A.

### Data procurement

For all identified studies, we will contact the corresponding author by email. If a current email address cannot be found or the author does not respond (up to three attempts), we will attempt to reach them by other means (phone, post, contact institution or any other available means of contact). Where IPD are available and authors or institutions are willing to share data, a data delivery agreement (DDA) will be drafted by both parties. A template DDA has been prepared for this study and it will be reasonably adapted if authors see the need to change it, after which it will be signed. Dutch ethics regulations do not require explicit ethical approval for conducting IPDMAs. However, where local ethics regulations require it, ethics approval will be sought before sharing data. Pseudonymised or anonymised data sets (all formats will be acceptable, eg, SPSS, Excel) and related data dictionaries will then be transferred and stored securely in a database at Erasmus MC, for use only as agreed on in the DDA. One original study investigator (first or senior author, at the discretion of the data owner) will be invited to be a co-author of the project if they are willing to assume responsibilities that meet authorship guidelines, as also stated in the DDA.

### Data processing and validation

We will convert all data sets to a common format, combine data sets with a new variable identifying the original trial and harmonise variables. Data checking will include evaluating baseline characteristics and results of comparisons for our main outcomes against results reported in original publications. We will also check to balance baseline participant characteristics in each treatment arm and evaluate the extent to which all randomised participants in the IPD datasets have been included in study analyses. Authors will be consulted in the case of any inconsistencies or discrepancies. In cases where discrepancies cannot be resolved, we will (on a case-by-case basis) either conduct a sensitivity analysis with that study removed or we will exclude the study from our analysis altogether.

Two independent investigators will parse data from all included published studies. From each study, we will extract the following data: country of study, funding source, study design, sample size, target population, inclusion/exclusion criteria, participant characteristics (age, sex, BMI, history of injury or surgery, comorbidities, medication use), type and context of intervention (eg, mesh placement technique, type of mesh, imaging techniques used to diagnose IH, suture technique), AAA characteristics, pain and quality of life pre–post as available. For all patient-reported outcomes, we will extract the recall period in addition to the outcome. Where IPD are available, we will conduct all analyses using IPD instead of aggregate data, following the data consistency checks described above.

### Study quality assessment

Two investigators will independently evaluate the risk of bias (ROB) for each included study using the Cochrane ROB tool, and disagreements will be resolved by a third investigator. Any authors involved in any included trial will not extract data from or assess the ROB in those trials. Duplicate publications will be identified to evaluate the trials and all available data simultaneously to maximise data extraction and correct bias assessment. The Cochrane ROB considers five domains of possible bias: randomisation, deviations from intended interventions, missing outcome data, measurement of the outcome and selection of the reported results. For each domain, ROB is rated as low, with some concerns or

high. The overall study will be considered to be of low ROB if all five domains are rated as low ROB, and high overall ROB if at least one domain is rated as high ROB or if some concerns are identified in multiple domains. We will consult the authors of the original publications in the event of inadequate reporting or inconsistencies. If indicated, we will email the authors to request data that may not have been sufficiently included in the primary publication.

The following trial-related data will be extracted:

► Trial characteristics: bias risk components, trial design, period and number of sites, countries where the trial was conducted, number of intervention arms, length of follow-up and inclusion and exclusion criteria.
► Participant characteristics and comorbidities: number of randomised participants, analysed participants, participants lost to follow-up, mean age, age range, sex ratio, specific patient-based inclusion criteria and treatment characteristics (eg, operating time).

## Assessment of heterogeneity

Forest plots will be constructed to visualise and assess any signs of heterogeneity. Statistical heterogeneity will be assessed using the $\chi^2$ test (threshold p<0.10), the quantities of heterogeneity will be measured with the $I^2$ statistic and possible heterogeneity will be assessed with relevant subgroup analyses.

All eligible patients from included RCTs will be included for final analysis if meeting the following criteria: adults (18 years or older) diagnosed with AAA using any common method (eg, radiographs, CT, clinical criteria, diagnosis by a healthcare professional). Additionally, inclusion criteria from included RCTs will be evaluated, and the criteria of the IPDMA will be amended if required. A potential subject who meets any of the following criteria will be excluded from the final analysis: emergency surgery or the presence of a mesh in the abdominal wall on the midline from previous hernia repair. Additionally, exclusion criteria from included RCTs will be evaluated, and the criteria of the IPDMA will be amended accordingly.

Sample size calculations stated in the included studies will be assessed. New power calculations will be performed for subgroup analyses that are performed on IPD. A one-stage IPDMA will be performed on the data received from the different included studies, which were identified through the literature search. We will conduct time-to-event analysis for all included patients using Cox regression analysis with trial and centre (nested under trial) as cluster terms to compare groups with and without the placement of the prophylactic mesh using the HR and the corresponding two-sided 95% CI.[20] Risk factors will be evaluated using Cox-regression analysis. Comparison of categorical and continuous variables between groups will be performed using mixed logistic regression analysis with, but not limited to, baseline value, age, gender and operation indication as possible covariates and trial and centre (nested under trial) as random effects.

## Missing data

To avoid bias induced by ignoring missing data in clinical research, it is widely acknowledged that imputation techniques can be considered to replace missing values. We anticipate that the proportion of missing values for the primary and secondary outcomes will be less than 5%, in trials that documented these parameters, and therefore, we will consider imputation. For partially missing data, traditional multiple imputation techniques will be performed per individual dataset, if not yet done by the researchers from the study but also if the proportion of missing values in relation to the total dataset is reasonably small allowing for the construction of a robust imputation model. However, in a secondary analysis, we will consider using multiple imputation and/or best-worst and worst-best-case scenarios if we can't ignore missing data. We will describe the proportion of missing values for each dataset included in the IPDMA.

## Treatment efficacy

To evaluate treatment efficacy, we want to employ a one-step MA on the primary outcome parameter, which is the IH rate. This will be evaluated using a time-to-event analysis. All data will be harmonised in one large dataset and analysed as pooled outcome data of all included patients in different RCTs, controlling for stratification per centre (indicated by an additional unique covariate for each of the different trials). We will analyse the effect of the treatment by intention to treat, regardless of the methods used in the original study. Cox regression analysis stratified per trial (on randomisation level) will be used to assess mesh efficacy for preventing IH occurrence. Effect sizes will be documented with relative risk (RR, 95% CI). For the secondary outcome measure postoperative complications within 30 days, such as SSI, SSO, fascial dehiscence and the need for re-operation (abdominal and other) at different time points during follow-up (eg, <30 days, 6 months, 1 year), we will conduct logistic regression models accounting for clustering on the trial level. Effect sizes will be documented with OR (95% CI). For the secondary outcome measure of pain and quality of life, we will use linear regression models accounting for clustering on the trial level as well and effect sizes will be documented with regression coefficients (β, 95% CI).

If a one-stage MA is not feasible, we will conduct a two-stage MA where we will first analyse each trial separately and then pool results across trials. In step 1, within each trial, we will evaluate the effect of assigned intervention by intention to treat, regardless of the method used in the original study. If study heterogeneity prevents us from harmonising data, then we will navigate this using a statistical approach based on available data. This will likely involve transforming data into standardised means differences or applying a proportion of maximum scaling methods.

In studies where we are unable to obtain IPD, we will extract aggregate data from published manuscripts as they are reported in the published articles. Similar

models will be performed for secondary outcomes as data permit. In cases of dichotomous outcomes, we will perform binary modelling and report effect sizes as RR (95% CI).

In step 2, we will perform random effects MA employing restricted maximum likelihood. We will report study heterogeneity as $I^2$ and $\tau^2$. In cases of notable heterogeneity ($I^2$>50%), we will consider possible sources such as study design, treatment duration, comparison treatment, treatment adherence or study quality. We will then consider performing meta-regression, subgroup analysis or sensitivity analyses to explain or account for these potential sources of heterogeneity. We will pool the results of studies both with and without IPD data after verifying that the effect sizes of IPD studies do not differ from non-IPD studies.

## Hypotheses

For the primary research question, it is hypothesised that prophylactic mesh reinforcement reduced the IH rate in comparison to primary sutured closure. Our secondary hypotheses are that postoperative complications such as SSI and SSO rate are comparable for the two methods of abdominal closure, while we hypothesise that prophylactic mesh reinforcement is superior regarding fascial dehiscence, pain, quality of life and the need for re-operation (abdominal-wall and other) as compared with the primary suture group.

## Treatment effect-modifier analyses

We will conduct treatment effect-modifier analyses to identify subgroups of individuals undergoing open-AAA surgery who benefit most from the placement of a prophylactic mesh by including interaction terms between the subgroup and treatment group in the corresponding regression analyses. We have proposed several subgroup characteristics that we hypothesise may modify the effect of the prophylactic mesh on our main outcome (IH formation), based on expert opinion. These proposed subgroups include the following baseline characteristics: (1) BMI score (patients with a higher BMI are at a higher risk for the development of an IH); (2) primary fascial closure with different suture length (SL)/wound length (WL) ratios (a higher SL to WL ratio results in fewer IHs, and therefore, the use of different SL/WL ratios might result in wrong conclusions and/or recommendations); and (3) patients with connective tissue disorders (can be associated with the formation of the AAA and also the healing of the abdominal wall and therefore the formation of an IH).

## Patient and public involvement

No patient involvement was sought for the development of the protocol for this IPDMA.

## Ethics and dissemination

No new data will be collected in this study. We will adhere to institutional, national and international regulations regarding the secure and confidential sharing of IPD, addressing ethics as indicated. We intend to publish the IPDMA in a peer-reviewed journal.

## Handling and storage of data and documents

Patient data from participating centres where the RCTs were held, will be anonymized and transferred via encrypted and secure data transfer. Before data transfer, a data delivery agreement will be signed by both parties. The EHS will handle and store data as an independent party. Only the assigned researcher in the Erasmus MC will have access to the data. No sponsor is present for the study.

**Author affiliations**
[1]Department of Surgery, Erasmus University Medical Centre, Rotterdam, The Netherlands
[2]Medicine, Erasmus University Rotterdam, Rotterdam, The Netherlands
[3]Department of Surgery, University Hospital of Ioannina, Ioannina, Greece
[4]Department of Vascular Surgery, University of Thessaly, Larissa, Greece
[5]Department of Vascular Surgery, North Bristol NHS Trust, Westbury on Trym, UK
[6]Department of Vascular Surgery, Gloucestershire Hospitals NHS Foundation Trust, Cheltenham, UK
[7]Department of Vascular Medicine, University Medical Center Hamburg-Eppendorf University Heart & Vascular Center, Eppendorf, Hamburg, Germany
[8]Department of Vascular Surgery, Hospital Robert Schuman Kirchberg Hospital, Luxembourg City, Luxembourg
[9]Department of General and Hepatobiliary Surgery and Liver Transplantation, Ghent University Hospital, Ghent, Belgium
[10]Department of Abdominal Surgery and Transplantation, Division of Abdominal Wall Surgery, CHU Liege, University of Liege, Liege, Belgium
[11]Department of Surgical Sciences, University of Genoa, Genoa, Italy
[12]Department of Surgery, AZ Maria Middelares Hospital, Ghent, Belgium

**Acknowledgements** We thank Dr. W.M. Bramer at the Erasmus MC, University Medical Centre, Medical Library for assisting us with developing our search terms and managing our search.

**Collaborators** European Hernia Society Prophylactic mesh study group collaborators: Holger Diener Department of Vascular Medicine, University Medical Center Hamburg-Eppendorf University Heart & Vascular Center, Eppendorf, Hamburg, Germany., Tilo Kölbel, Department of Vascular Medicine, University Medical Center Hamburg-Eppendorf University Heart & Vascular Center, Eppendorf, Hamburg, Germany. Wolfgang Reinpold Hamburg Hernia Centre, Department of Hernia and Abdominal Wall Surgery, Helios Mariahilf Hospital Hamburg, Teaching Hospital of the University of Hamburg, Hamburg, Germany. Antonia Zapf Institute of Medical Biometry and Epidemiology, University Medical Center Hamburg-Eppendorf, Hamburg, Germany. Eric Bibiza-Freiwald Institute of Medical Biometry and Epidemiology, University Medical Center Hamburg-Eppendorf, Hamburg, Germany. Vaia K. Georvasili Department of Surgery, University Hospital of Ioannina, Ioannina, Greece.

**Contributors** All authors were involved in the study design and all will contribute to the interpretation of the results. RvdB contacted the potential data deliverers, will coordinate the data collection and perform/ supervise the data analyses. RvdB wrote the manuscript together with FdH and PT. RvdB and FdH will have full access to the study data. RvdB, FdH, CB, CS, MM, PMB, JJE, EB, SH, FB, OD, DS, FM, PT and the European Hernia Society Prophylactic Mesh Study Group Collaborators approved the final manuscript.

**Funding** The authors have not declared a specific grant for this research from any funding agency in the public, commercial or not-for-profit sectors.

**Competing interests** Participating authors of the I-PREVENT study have previously worked on the PRIMA and PRIMAAT trials. These authors will be excluded from doing quality evaluation and data extraction of relevant trials. No further statements of competing interests need to be disclosed.

**Patient and public involvement** Patients and/or the public were not involved in the design, conduct, reporting or dissemination plans of this research.

**Patient consent for publication** Not applicable.

**Provenance and peer review** Not commissioned; externally peer reviewed.

**ORCID iD**
Rudolf van den Berg http://orcid.org/0009-0001-8850-6166

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
