## [Reviewer comments · BMJ Open]

ARTICLE DETAILS

TITLE (PROVISIONAL)	Protocol for an Independent Patient Data Meta-Analysis of Prophylactic Mesh Placement for Incisional Hernia Prevention (after Abdominal Aortic Aneurysm surgery): A collaborative European Hernia Society project
AUTHORS	van den Berg, Rudolf; den Hartog, Floris; Bali, Christina; Matsagkas, Miltiadis; Bevis, Paul; Earnshaw, Jonothan; Debus, Eike; Honig, Susanne; Berrevoet, Frederik; Detry, Olivier; Stabilini, Cesare; Muysoms, Filip; Tanis, Pieter; Prophylactic Mesh Study Group Collaborators, European Hernia Society

VERSION 1 – REVIEW

REVIEWER	Huang, Li-Ching Vanderbilt University, Biostatistics
REVIEW RETURNED	13-Nov-2023

GENERAL COMMENTS	1. The primary aim is to evaluate the efficacy and effectiveness of the use of prophylactic mesh for patients who underwent post open abdominal aortic aneurysm surgery. What's outcome to measure the efficacy and effectiveness of the use of prophylactic mesh?2. The secondary aims include evaluation of postoperative complications, pain, and quality of life. Please provide the clear definitions for each secondary outcome. That is, at which time point and how the postoperative complication, pain and quality of life will be measured or defined.3. Page 5, line 23. The authors say the primary outcome is the time to incisional hernia occurrence during long-term follow-up. Please define "long-term follow-up".4. Page 8, the section of treatment efficacy. I think this is mainly for studying the primary aim. Again, it is lack of detail of outcome. The authors mentioned they will report effect sized as relative risk with 95% CI. I assume the outcome is not time to long-term incisional hernia occurrence.5. Please lay out the hypotheses for each study aims (primary and each secondary).6. Please describe statistical analysis approaches that will be used to address for each hypothesis (primary and each secondary) in sufficient detail.
--

REVIEWER	López-Plaza, José Antonio Ramón y Cajal University Hospital, urology
REVIEW RETURNED	17-Jan-2024

GENERAL COMMENTS	I think it is a very good idea to provide more scientific evidence for laparotomy wall closure techniques. As a tip, I think that certain technical characteristics such as mesh
---

	placement technique, type of mesh implanted, imaging techniques used for the diagnosis of incisional hernia, etc. should be included.
REVIEWER	Ulutas, M.E. University of Health Sciences, General Surgery
REVIEW RETURNED	18-Jan-2024
GENERAL COMMENTS	It is really valuable that you evaluate individual participant data (IPD) in your study, rather than the results obtained from published articles. Congratulations to the authors.

VERSION 1 – AUTHOR RESPONSE

Reviewer: 1
Dr. Li-Ching Huang, Vanderbilt University

Comments to the Author:

Author: We would first and foremost like to thank the reviewer for the clear comments on our manuscript.

1. The primary aim is to evaluate the efficacy and effectiveness of the use of prophylactic mesh for patients who underwent post open abdominal aortic aneurysm surgery. What's outcome to measure the efficacy and effectiveness of the use of prophylactic mesh?

Author response: The primary outcome parameter is incisional hernia rate, which we intend to determine at 2, 3 and 5 years after surgery. We have now more clearly formulated our primary outcome parameter in the paragraph 'Aims'.

2. The secondary aims include evaluation of postoperative complications, pain, and quality of life. Please provide the clear definitions for each secondary outcome. That is, at which time point and how the postoperative complication, pain and quality of life will be measured or defined.

Author response: We added clear outcome measures for the secondary outcomes, such as noting the types of rating scales that might have been used in the original trials for assessing the pain and quality of life of the patients. The time points at which we want to evaluate these secondary outcomes are 30 days, 6 months and 1 year postoperatively, which has been added to the revised manuscript in the paragraph 'Aims'. Of course, this will also depend on the availability of specific data on different postoperative time points in each of the included trials.

3. Page 5, line 23. The authors say the primary outcome is the time to incisional hernia occurrence during long-term follow-up. Please define "long-term follow-up".

Author response: We have extended the 'Aims' paragraph to include that we want to assess incisional hernia rate at 2, 3, and 5 years postoperatively. Again, this will also depend on the availability of these data in the included trials.

4. Page 8, the section of treatment efficacy. I think this is mainly for studying the primary aim. Again, it is lack of detail of outcome. The authors mentioned they will report effect sized as relative risk with 95% CI. I assume the outcome is not time to long-term incisional hernia occurrence.

Author response: Indeed, treatment efficacy will be determined using the primary outcome measure, which is IH rate. We removed "time to long-term IH occurrence" from the manuscript, because the reviewer is correct that this is not the outcome of interest. We added more explanation in this section. Effect sizes from the Cox-regression models accounting for clustering on trial level, will be described with RRs.

5. Please lay out the hypotheses for each study aims (primary and each secondary).

6. Please describe statistical analysis approaches that will be used to address for each hypothesis (primary and each secondary) in sufficient detail.

Author response: We added hypothesis (5) for the primary and secondary outcomes under the “hypothesis” section and described the fitting statistical analysis (6) for each of the hypothesis.

Reviewer: 2

Dr. José Antonio López-Plaza, Ramón y Cajal University Hospital

Comments to the Author:

I think it is a very good idea to provide more scientific evidence for laparotomy wall closure techniques.

As a tip, I think that certain technical characteristics such as mesh placement technique, type of mesh implanted, imaging techniques used for the diagnosis of incisional hernia, etc. should be included.

Author response: We would very much like to thank the reviewer for the positive comments. We agree with the reviewer that these technical characteristics should be considered and added these parameters in the “Data processing and validation” section.

Reviewer: 3

Dr. M.E. Ulutas, University of Health Sciences

Comments to the Author:

It is really valuable that you evaluate individual participant data (IPD) in your study, rather than the results obtained from published articles. Congratulations to the authors.

Author response: We would like to thank this reviewer for the positive comments.

VERSION 2 – REVIEW

REVIEWER	Huang, Li-Ching Vanderbilt University, Biostatistics
REVIEW RETURNED	04-Mar-2024
GENERAL COMMENTS	The authors have responded to all my questions. I have no further concerns that require another revision.